# ThermoRL: Structure-Aware RL for Protein Mutation Design to Enhance Thermostability

## Abstract

Designing mutations to optimize protein thermostability remains challenging due to the complex relationship between sequence variations, structural dynamics, and thermostability, often assessed by $\Delta\Delta G$ (the change in free energy of unfolding). Existing methods rely on experimental random mutagenesis or one-shot predictions over fixed mutation libraries, which limits design space exploration and lacks iterative refinement capabilities. We present **ThermoRL**, a reinforcement learning (RL) framework that integrates graph neural networks (GNN) with hierarchical Q-learning to sequentially design thermostabilizing mutations. ThermoRL combines a pre-trained GNN-based encoder with a hierarchical Q-learning network and employs a surrogate model for reward feedback, to guide the agent in selecting both mutation positions and amino acid substitutions. Experimental results show that ThermoRL achieves higher or comparable rewards than baselines while maintaining computational efficiency. It effectively avoids destabilizing mutations, recovers experimentally validated stabilizing variants, and generalizes to unseen proteins by identifying context-dependent mutation sites. These results highlight ThermoRL as a scalable, structure-informed framework for adaptive and transferable protein design.

## 1 Introduction

Proteins are a diverse and valuable group of molecules that play important roles in many clinical, industrial, and research applications Gurung et al. (2013); Jemli et al. (2016); Maghraby et al. (2023). Thermodynamic stability is a key property of proteins, as naturally evolved proteins are often only marginally stable under normal conditions Goldenzweig & Fleishman (2018). Therefore it is a critical property for industrial Hammond (2007); Nezhad et al. (2022) and biomedical applications Shimanovich et al. (2014); Demetzos & Pippa (2019).

The thermodynamic stability of a protein is generally quantified by its Gibbs free energy ($\Delta G$) during the folding process Lazaridis & Karplus (2002); Chong & Ham (2014); Ahmad (2022). The magnitude of $\Delta G$ is determined by interactions among amino acid residues and between the protein and its surrounding biophysical environment. When a mutation introduces an amino acid substitution, the stability of the mutant protein typically changes, and this difference, $\Delta\Delta G$, is expressed as the change in Gibbs free energy relative to the wild-type protein. While the stability of naturally evolved enzymes is sufficient for biological processes, industrial applications such as biocatalysis and biofuel production often demand enhanced stability Bommarius & Paye (2013); Borrelli & Trono (2015); Bell et al. (2021). Identifying advantageous point mutations that improve protein stability is crucial for advancing research and biocatalysis, enabling broader and more efficient industrial applications.

However, selecting beneficial mutations remains a major challenge due to the vast mutational search space and the complex relationship between sequence, structure, and stability. Although deep learning has brought transformative advances to protein stability engineering, we argue that *existing mutation optimization methods still face critical limitations* when applied to thermostability improvement, due to two major **challenges**: (1) **Inefficient Identification of Optimal Mutation Sites**. Traditional directed evolution (DE) (Figure 1A) relies on random mutagenesis across the entire sequence, with each round of evolution testing all possible mutations at one position to identify the optimal amino acid Xiong et al. (2021); Wang et al. (2021), which is labor-intensive and inefficient. ML-assisted approaches (Figure 1B) replace experimental screening with in silico scoring using supervised models,

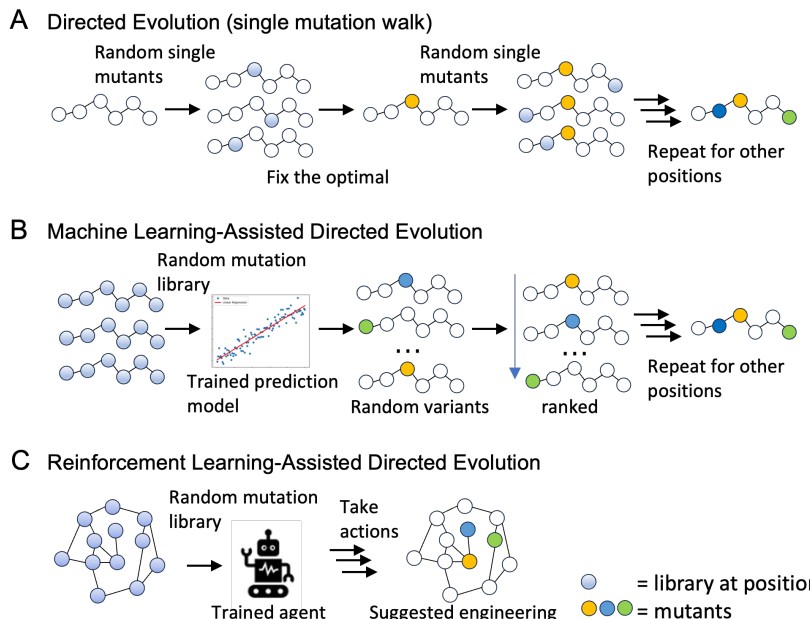

Figure 1: Comparative strategies for protein engineering. (A) Directed evolution begins with a single protein sequence, introducing random mutations and iteratively testing them to identify the optimal amino acid configurations. (B) Machine learning-assisted directed evolution utilizes protein sequence dataset and applies ML models to predict and prioritize the performance of mutations in a predesigned library. (C) Our approach **ThermoRL**: Our RL-assisted approach utilizes structural and sequential-level protein datasets to train an agent that can recommend optimal mutation strategies for protein engineering.

but often follow a "predict-then-rank" pipeline that requires pre-generating large mutation libraries. Jumper et al. (2021); Baek et al. (2021); Pancotti et al. (2022). (2) **Limited Use of Structural Context in Design Space.** While structural information has been incorporated into predicting mutation effects, as demonstrated by models like ThermoMPNN Dieckhaus et al. (2024), approaches by Li et al. (2020) and Wang et al. (2023a), optimization frameworks for protein design do not directly integrate structural cues into their decision-making processes. Instead, structure is used post hoc to score or filter candidates. Furthermore, many RL-based frameworks operate at the sequence level and are not designed for generalized thermostability optimization across proteins with diverse folds Angermueller et al. (2019); Wang et al. (2023b).

To address these challenges, we propose a novel structure-aware hierarchical reinforcement learning (HRL) framework, named ThermoRL, which integrates 3D structural information into mutation decision-making (Figure 1C). To optimally select the mutation sites, ThermoRL formulates the mutation selection process as a Markov Decision Process (MDP) and leverages hierarchical Q-learning networks Ho et al. (2006); Pateria et al. (2021) to solve it. This approach reduces the search space, eliminating the need for exhaustive evaluation of all possible mutations and improving computational efficiency. By incorporating 3D structure through graph neural network (GNN) representations, the agent learns structure-aware mutation policies that generalize across diverse protein topologies.

While reinforcement learning (RL) has achieved remarkable success in various high-dimensional decision-making tasks, most existing protein optimization frameworks either focus on predicting mutation effects from fixed candidate sets or require training separate models for individual proteins. Unlike models such as those benchmarked in ProteinGym Notin et al. (2023) which focus on the prediction of the zero-shot mutation effect, our approach focuses on learning generalizable mutation strategies through sequential decision-making. Unlike previous $\Delta\Delta G$ prediction models or single-target optimization, ThermoRL learns a sequential policy to propose mutation sites and substitutions informed by protein 3D structure. This enables trajectory-level planning over multiple residues, reducing the combinatorial explosion and promoting targeted design across di-

verse proteins.Overall, in this paper, we highlight our **contributions** as follows: (i) **Generalizable structure-informed optimization and transferable across unseen proteins.** ThermoRL is the first reinforcement learning framework for thermostability optimization that combines GNN-based protein encodings with hierarchical decision-making, enabling transferable design across unseen proteins. It is not a single-target model that requires training for different protein target. (ii) **Trajectory-based mutation planning.** Unlike predict-then-rank baselines, ThermoRL formulates mutation design as a sequential decision process, allowing the model to actively navigate the mutation landscape using learned structure-aware policies. (iii) **Improved efficiency through hierarchical control.** The hierarchical policy significantly reduces the mutation action space, enabling efficient learning and faster convergence compared to flat or exhaustive strategies.

## 2 RELATED WORK

**Traditional Strategies for Point Mutation** Traditional strategies to enhance enzyme thermal stability and activity Xiong et al. (2021); Rahban et al. (2022) include directed evolution Madhavan et al. (2021); Li et al. (2024), rational design Du et al. (2024); Qu et al. (2022); Reetz (2022), and semirational design Nezhad et al. (2023) . However, traditional methods struggle to explore the vast sequence space effectively and rely heavily on prior chemical knowledge. In silico approaches address these limitations by predicting the effects of point mutations on protein stability, using empirical energy functions to model covalent and noncovalent atomic interactions for mutation evaluation Alford et al. (2017); Krishna et al. (2024), or sequence-derived information, such as position-specific substitution matrices Bednar et al. (2015).

**Deep-Learning Prediction Models for $\Delta\Delta G$** Recently, deep learning models trained on large-scale mutation datasets have achieved notable success. Some methods rely solely on sequence-based features, using machine learning and statistical models like PoPMuSiC 2.1 Dehouck et al. (2011), CUPSAT Parthiban et al. (2006), STRUM Quan et al. (2016), and DeepDDG Cao et al. (2019) to infer stability changes from amino acid sequences and evolutionary data. Others incorporate structural information; for instance, ThermoNet Li et al. (2020) and RaSP Blaabjerg et al. (2023) use 3D CNNs to process voxelized protein structures, effectively capturing spatial interactions but at a high computational cost. Graph GNNs, such as ProS-GNN Wang et al. (2023a), BayeStab Wang et al. (2022), and ThermoMPNN Dieckhaus et al. (2024), efficiently represent proteins as graphs, leveraging transfer learning from ProteinMPNN Dauparas et al. (2022).

**Application of Reinforcement Learning** Deep reinforcement learning (DRL) has achieved significant advances in solving sequential decision-making and automated control tasks, with applications in areas such as self-driving cars Jaritz et al. (2018); Spielberg et al. (2019), robot control Singh et al. (2022); Liu et al. (2021), and AI for games Lample & Chaplot (2017); Shao et al. (2019). More recently, DRL has been applied to structured graph data, such as a GNN-based policy network for robotic control proposed by Wang et al. (2018). In chemistry and molecular graph mining, DRL has been applied to generate molecular graphs and predict reaction outcomes You et al. (2018); Do et al. (2019). In design new proteins from a target protein for the specific function, DyNA-PPO Angermueller et al. (2019) employs model-based RL with PPO and adaptive agent selection, but sparse rewards and sequence misfold risks arise as designs are evaluated only at the episode's end; EvoPlay Wang et al. (2023b) uses a policy-value network to optimize protein sequences via single-residue mutations, focusing solely on 1D sequences while incurring high computational costs due to its binary matrix action space. In this paper, we propose an HRL approach for protein design, targeting thermostability—a universal objective across all proteins. Our model addresses the limitations of existing methods by efficiently identifying optimal mutation sites while incorporating protein structural information, and generalizing to unseen proteins.

## 3 THERMORL FRAMEWORK

### 3.1 GRAPH EMBEDDING

**Graph Representation:** The protein structures were represented by contact map graphs. The $i$-th protein can be represented by graph $\mathcal{G}_i = (V_i, E_i)$, where $V_i = \{v_j^{(i)}\}_{j=1}^{|V_i|}$ is the set of amino acid residues, and edges $E_i = \{(v_j, v_k) \mid v_j, v_k \in V_i\}$ represent interactions between residues. In these

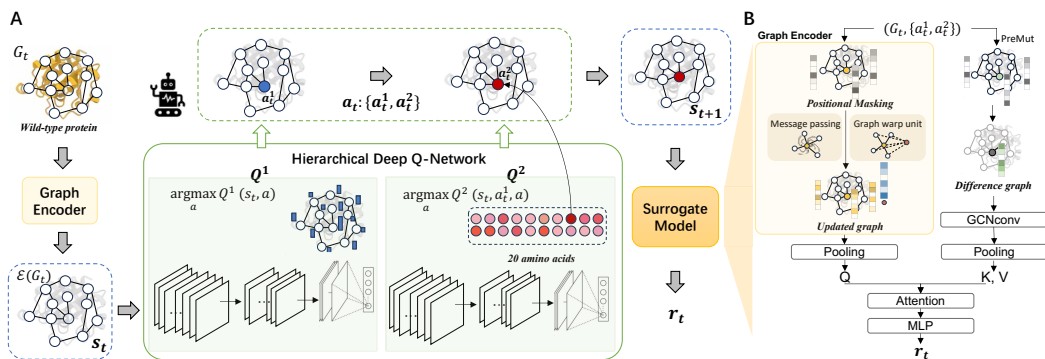

Figure 2: Overview of the two-stage ThermoRL framework, where (A) the agent sequentially selects mutation positions and amino acid substitutions to optimize protein stability, and (B) the surrogate model estimates the stability impact of a given mutation, $\Delta\Delta G$, as the reward signal for the HRL agent, leveraging a pretrained graph encoder to extract structural features.

protein structure graphs, nodes $V$ represent the sequence indices of individual amino acids, each of which has associated features $x(v_j^{(i)})$, derived from a combination of a one-hot encoding of the 20 amino acid types and five key physicochemical properties: molecular weight, $pK_a$, $pK_b$, $pK_x$, and $pI$. These features offer detailed insights into each amino acid's mass, acid and base dissociation constants, and isoelectric point, all of which are crucial for understanding their roles in protein structures and interactions. Edges between nodes represent spatial connections determined by the three-dimensional coordinates of the $\alpha$-Carbon atoms, which are the central carbon atoms in the backbones of each residue. A predefined cut-off distance of 8 Å is used to establish these connections, ensuring that only residues within this distance are considered neighbours Wang et al. (2017; 2024).

**Graph Encoder:** The graph encoder, defined as $\mathcal{E}$, utilizes a multi-head GNN to extract graph representations from protein structures, integrating positional encodings to retain sequential context. The features are updated iteratively using the message passing mechanism. Additionally, a global super-node representation is constructed by aggregating node features and dynamically refined via a GRU-based update mechanism. Therefore, the final embedded protein graph for $G$ can be represented as $\mathcal{E}(G)$, enriched with structural and relational information, serving as input for downstream tasks.

## 3.2 REINFORCEMENT LEARNING ENVIRONMENT SETTING

Given an input graph $G_i = (V_i, E_i)$, we model the protein mutation optimization problem as a Finite Markov Decision Process (MDP) $(\mathcal{S}, \mathcal{A}, \mathcal{P}, \mathcal{R}, \gamma)$, which is defined by the state space $\mathcal{S}$, action space $\mathcal{A}$, transition probability $\mathcal{P}$, reward $\mathcal{R}$, and discount factor $\gamma$. Specifically, the MDP environment in this paper is defined as follows:

**State:** At time $t$, the state $s_t$ is represented as the embedded protein graph $\mathcal{E}(G_t)$, derived from the pre-trained graph encoder. The RL state is defined over a fixed node set $V_i$, where the agent selects a specific node $v_i^j$ for mutation.

**Action:** In the protein mutation environment, the agent's goal is to determine the optimal mutation by (i) selecting the mutation site from the protein graph's node set $V$ (where to mutate), and (ii) choosing a replacement amino acid from the set $C$ with size $|C| = 19$ (which amino acid to mutate into, excluding the wild-type residue). Therefore, a single action at time step $t$ is defined as $a_t \in \mathcal{A} \subseteq V \times C$. In traditional RL approaches designed for single-protein optimization Wang et al. (2023b), the action space is $O(|V||C|)$, which becomes computationally expensive when scaling (or transfer) to large datasets with multiple proteins to train a universal agent. To address this, we propose a hierarchical action strategy that reduces the action space and enables more efficient exploration. As illustrated in Figure 2 (A), at time $t$, the agent performs two actions: It first selects the protein position to be mutated ($a_t^1$) from the node set $V_i$ of the input protein graph $G_i$. Next, an action ($a_t^2$) is taken to select the replacement amino acid from the substitution subset $C_i$. The combined action is represented as $\mathbf{a_t} = \{a_t^1, a_t^2\}$. After performing $a_t^1$ and $a_t^2$, the mutation information is passed to the surrogate

module for reward calculation. By introducing this hierarchical action strategy, the exploring action space is reduced from $O(|V||C|)$ to $O(|V| + |C|)$, significantly improving computational efficiency.

**Reward:** The reward function guides the agent in identifying optimal mutations to enhance thermostability. However, the scarcity of experimental data limits reward signals, constraining design space and hindering exploration. To address this, we introduce a **surrogate model**, with structure shown in Figure 2(B), that predicts the reward signals, $\Delta\Delta G$, based on the wild-type protein and mutation information. The surrogate model incorporates two key components: the wild-type graph and a difference graph. The wild-type graph captures the essential structural information of the original protein through a graph encoder. Mutation information is represented by the difference graph, constructed by aligning the wild-type structure with the mutant structure predicted by PreMut Mahmud et al. (2023). A unique aspect of our surrogate model is the integration of cross-attention to directly map the difference graph's mutation-specific features to the wild-type graph embeddings. Unlike conventional approaches that aggregate graphs as a whole, our method retains residue-level granularity and explicitly models mutation-induced interactions. This is followed by a multilayer perceptron (MLP) that predicts the mutation effect score as the reward value.

**Definition 3.1.** (Surrogate Model) Given a protein graph $G_i$, node $v_j^i$ is mutated with an amino acid $c_j \in C$. The surrogate model $\mathbb{S}$ predicts the $\Delta\Delta G$ as:

$$\Delta\Delta G = \mathbb{S}(\mathcal{E}\{G_i\}, v_j, c_j) \tag{1}$$

where $\mathcal{E}(G_i)$ is the embedded graph.

Therefore, at each time step $t$, the reward is defined as:

$$r(s_t, \mathbf{a_t}) = r(G_t, \{a_t^1, a_t^2\}) = \mathbb{S}(\mathcal{E}\{G_t\}, a_t^1, a_t^2) \tag{2}$$

where $\mathcal{E}(G_t)$ is the graph embedding generated by the pre-trained GNN encoder, and $\{a_t^1, a_t^2\}$ corresponds to the hierarchical actions selecting a mutation node and its corresponding amino acid substitution.

**Terminate:** The agent terminates after determining a mutant amino acid for the selected position, based on the learned Q-values. For multiple mutations, the maximum number of hierarchical selection steps can be increased, or a reward threshold set to stop exploration.

### 3.3 HIERARCHICAL DEEP Q NETWORK

Q-learning is an off-policy optimization method that identifies an optimal policy by maximizing the expected total reward over all future steps starting from the current state. It achieves this by directly solving the Bellman optimality equation, as shown below:

$$Q(s_t, a_t) = r(s_t, a_t) + \gamma \max_a Q(s_{t+1}, a). \tag{3}$$

where $Q(s_t, a_t)$ represents the Q-value for $(s_t, a_t)$, $r$ is the immediate reward, and $\gamma$ is the discount factor that determines the importance of future rewards. The action $a_t$ s selected using the greedy policy $\pi(s_t) = \arg\max_a Q(s_t, a)$. Deep Q learning Networks (DQN), extends the basic Q-Learning algorithm by utilizing deep neural networks to approximate Q-values. It introduces a target network which is the secondary neural network, to compute the target Q-values. This target network is updated less often than the primary network during the learning process, which can be used as the guide network for the primary network. The Q-learning loss function is:

$$\mathbb{E}_{(s,a,s',r)\sim\mathcal{M}} \left[ \left( r + \gamma \max_{a'} Q_t \left( s', a' \mid \theta^- \right) - Q(s, a \mid \theta) \right)^2 \right] \tag{4}$$

where $Q_t$ is the target action-value function, and its parameters $\theta^-$ are updated with $\theta$ every h steps, $\mathcal{M}$ represents the history replay buffer.

In this paper, we adopt a hierarchical Q-learning network that performs hierarchical actions $a_t = \{a_t^1, a_t^2\}$ instead of a single action, where $a_t^1 \in V$ and $a_t^2 \in C'$. To perform the hirechical actions, we integrate two DQNs to model the $Q$ values over the actions as $Q = \{Q^1, Q^2\}$. The first action $a_t^1$ is performed by the policy guided by the first DQN $Q^1$. Consequently, the fist action is determined by the optimal action-value function $Q^1$ based on the greedy policy:

$$a_t^1 = \arg\max_{a \in V} Q^1(s_t, a, \theta_1)$$

where $\theta_1$ represents the trainable weights of $Q^1$. With the first protein position selected by taking the first action, the RL agent then takes the second action hierarchically to replace the amino acid for the selected protein position:

$$a_t^2 = \arg\max_{a \in C'} Q^2(s_t, a_t^1, \theta_2) \tag{5}$$

where $\theta_2$ is the trainable weights for $Q^2$. In general, with the proposed hierarchical DQN $\{Q^1, Q^2\}$, ThermoRL integrates hierarchical action value functions to model Q values over hierarchical actions $\{a^1, a^2\}$.

### 3.4 Training Algorithm

The surrogate model leverages a pre-trained graph encoder, initially trained in an unsupervised manner on a diverse set of protein structures. During surrogate training on task-specific datasets (thermostability $\Delta\Delta G$), the graph encoder was fine-tuned with a reduced learning rate to adapt to the regression task. Performance was evaluated using root-mean-square error ($RMSE$) and $R^2$, which measured prediction accuracy and model generalization, respectively. A robust 5-fold cross-validation strategy was employed to ensure reliable performance estimates by averaging results across multiple partitions. The fine-tuned graph encoder generates structural embeddings for the RL agent, facilitating efficient mutation optimization. The surrogate model serves as the reward signal in the RL framework.

To train the RL agent, DQN uses a replay buffer to store experiences comprising the state, action, reward, and next state. The network trains on randomly sampled mini-batches from this buffer, which enhances sample efficiency. Similarly, for hierarchical DQNs, this method is applied by simulating the selection process to generate training data, represented as batches of experiences $\mathcal{B} = (s, \{a^1, a^2\}, r, s')$, which are stored in a memory buffer $\mathcal{M}$. During the training phase, batches of experiences $B$ are drawn uniformly from $\mathcal{M}$. The training process uses a loss function defined in Eq. 4, and an $\epsilon$-greedy policy is employed, where the agent takes random actions with a probability of $\epsilon$. In the proposed model, all trainable parameters $\theta$ for the action-value functions $Q_1, Q_2$ are implemented using two-layer multi-layer networks. The detailed algorithm is presented in Appendix.

## 4 Experiments

### 4.1 Experimental Setup

**Dataset** We utilized four datasets for this study: Megascale and Largescale for training, and Ssym and S669 for unseen testing. The training datasets represent the most comprehensive collections available, providing extensive mutation coverage and structural diversity crucial for robust model training. Megascale Tsuboyama et al. (2023) integrates high-throughput protease sensitivity experiments, focusing on small proteins ($< 75$ residues) with dense mutation coverage, making it an ideal resource for training mutation prediction models. Largescale, created by merging Q3421 Quan et al. (2016) and Q5440 Cao et al. (2019), underwent rigorous cleaning to remove duplicates and errors. It spans a broader range of protein lengths but has sparse mutation coverage. Figure 3A compares the mutation count and length distributions across these datasets. The testing datasets were selected based on their widespread use as benchmarking standards, ensuring fair and consistent evaluation against prior studies. Ssym Li et al. (2020) includes single mutations within well-characterized crystal structures, while S669 Pancotti et al. (2022) comprises mutations in proteins selected for sequence dissimilarity from training datasets. Figure 3B compares the biological origins, showing a higher proportion of artificial proteins in Megascale compared to other datasets. Structural data were extracted from PDB files to construct molecular graphs, enabling the model to capture structural and relational features. We also conducted a structural and embedding-based similarity analysis between training and test datasets Appendix 6.3.

**Pre-trained Surrogate Model** We trained two surrogate models, MegaSurrogate and LargeSurrogate, using the Megascale and Largescale datasets, respectively. Thermostability prediction was evaluated using Root Mean Squared Error (RMSE), Coefficient of Determination ($R^2$), and Pearson Correlation Coefficient (PCC). On training data, MegaSurrogate achieved stable performance for shorter proteins, likely due to the dense mutation coverage in Megascale, whereas LargeSurrogate showed greater variability, reflecting the sparse and diverse nature of Largescale (Appendix Figure 7). For

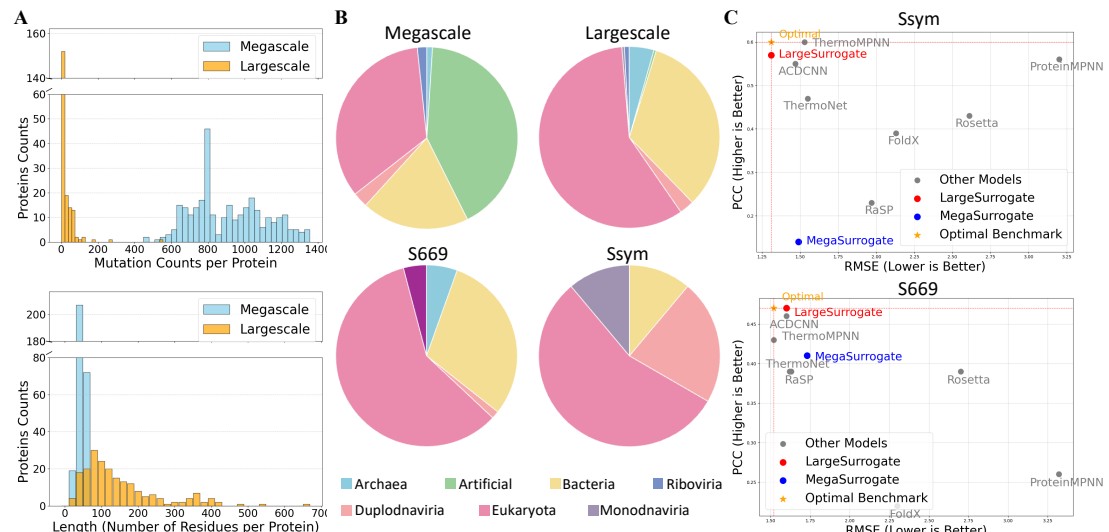

Figure 3: (A) Distribution analysis of experimental datasets, with the upper panel showing mutation counts per protein and the lower panel protein lengths for the training datasets (Megascale, Largescale). (B) The proportional composition of biological origins in the protein training datasets and unseen testing datasets (S669, Ssym) for the ThermoRL model. (C) Comparison on unseen test datasets (S669, Ssym) between the surrogate models and baseline models using RMSE and PCC metrics.

unseen testing, the surrogate models were evaluated on the widely used Ssym and S669 benchmark datasets. As shown in Figure 3C), both surrogate models achieved strong accuracy on S669, with LargeSurrogate also outperforming on Ssym, likely due to its broader training coverage and stronger generalization. We therefore selected LargeSurrogate as the reward estimator in ThermoRL. Figure 3C also includes supervised baselines such as ThermoMPNN Dieckhaus et al. (2024) , which predict mutation effects in a "predict-then-rank" manner over fixed candidate sets. These models do not perform decision-making or mutation selection, and are thus not directly comparable to ThermoRL. We include them here solely for benchmarking $\Delta\Delta G$ prediction accuracy. Therefore, they are not directly comparable to ThermoRL, but are included as surrogate baselines to benchmark the accuracy of the prediction of $\Delta\Delta G$, shown in Figure 3C. Additionally, we provide expanded baseline results in Appendix 8.1, covering ThermoNet, ThermoMPNN, RaSP, ProS-GNN, Rosetta, FoldX, and ACDCNN. These additional results confirm that our surrogate achieves state-of-the-art $\Delta\Delta G$ prediction accuracy compared to both learning-based and physics-based baselines.

**Joint Probability** To quantify the performance of the trained ThermoRL and enable comparisons with the results from surrogate models and experiments, we calculated the joint probability with two steps. The site-level probability $P(position)$ is obtained by applying a softmax over all position-wise rewards. The conditional probability $P(mutation \mid position)$ is computed using softmax over mutation rewards at a given position. The final joint probability is defined as $P(position) \cdot P(mutation \mid position)$.

**Baselines and Model Variants** We compare ThermoRL against three heuristic-based optimization methods designed for protein mutation: BO-GP, BO-ENN, and Random Search. Bayesian Optimization (BO) Hie et al. (2020) is a global optimization technique that approximates expensive objective functions using a surrogate model and selects query points by balancing exploration and exploitation. BO-GP employs a Gaussian Process as the surrogate model, while BO-ENN replaces it with an ensemble of neural networks to better capture non-linear mutation effects. Random Search serves as a reference baseline, generating sequences uniformly across the search space. All methods were evaluated by average cumulative reward on both individual proteins and a larger set of 100 protein samples. Each experiment was repeated 10 times, and the average cumulative reward was reported. We also measured computational efficiency in terms of time required to obtain optimal results. To assess the contribution of architectural components, we conducted ablation studies (Appendix Figure 9), comparing the full model with a variant that excludes the graph encoder. The results show a marked performance drop, confirming the importance of structural embeddings for generalization.

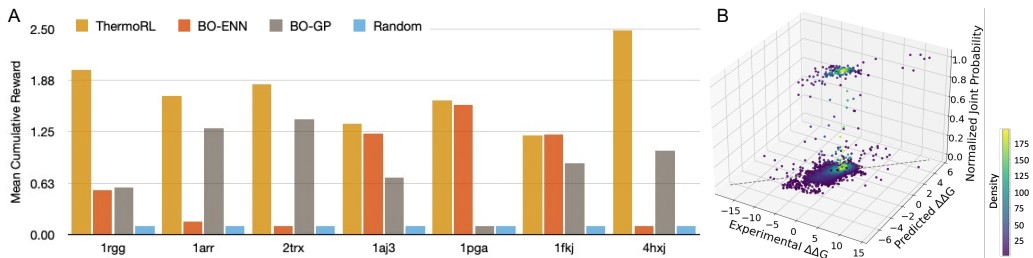

Figure 4: ThermoRL performance on large-scale protein datasets. (A) Comparison of cumulative rewards across a representative subset of proteins, showing that ThermoRL outperforms heuristic baselines (BO-ENN, BO-GP, and Random Search). (B) Visualization of the joint distribution of experimental and surrogate-predicted $\Delta\Delta G$ values. Points with higher vertical elevation correspond to mutations prioritized by the RL policy, illustrating ThermoRL's preference for stabilizing mutations while avoiding destabilizing ones. High-probability points "float" above the enumerated mutation library, demonstrating trajectory-level planning and robust policy generalization.

## 4.2 THE EFFECTIVENESS OF THERMORL

In this section, we show the ThermoRL performance on training Largescale dataset.

**Comparison with Baseline models** As shown in the experimental results in Figure 4A, ThermoRL achieved higher or comparable cumulative rewards relative to BO-based and random search baselines across selected proteins from Largescale. The results indicate that ThermoRL achieves rewards that are either higher than or comparable to those obtained by BO-based optimization methods. Notably, while BO and RS methods are specifically designed to identify the optimal protein mutation for each individual protein, ThermoRL is trained across a broader dataset, enabling generalizable policies. These findings emphasize ThermoRL's strong generalization capabilities across diverse proteins.

**Visualization on Training Dataset** We evaluated the ThermoRL's overall performance on the full dataset by visualizing the relationship between experimental $\Delta\Delta G$, surrogate model predictions for $\Delta\Delta G$, and the normalized joint probability in Figure 4B. High joint probability values, computed using position and mutation rewards, provides a comparable metric across proteins in same dataset, indicate mutations prioritized by the RL agent, "floating" above the rest. The visualization reveals that high-probability points are largely absent in regions with negative $\Delta\Delta G$ values, indicating the agent avoids destabilizing mutations. A small cluster of high-probability points appears where $\Delta\Delta G$ is slightly positive, reflecting a preference for mutations with mild stability improvements. In regions where both experimental and predicted $\Delta\Delta G$ values are near zero, a high density of data points is observed, making it difficult to distinguish individual mutation designs. ThermoRL prioritizes feasible and stable designs by selecting mutations with minimal differences between experimental and predicted $\Delta\Delta G$ values. Its preference for slightly positive $\Delta\Delta G$ further suggests an inclination toward incremental improvements in protein stability while avoiding high-risk changes.

## 4.3 THE GENERALISATION AND TRANSFERABILITY OF THERMORL

We assessed ThermoRL's generalization capability by measuring its average performance across 200 unseen proteins and, to gain mechanistic insights, conducted detailed case studies on three representative examples with PDB ID 1J8I, 2PR5, and 2WQG.

**Comparision with Baseline models** We evaluated ThermoRL and baseline methods (BO-GP, BO-ENN, and Random Search) across the 200 unseen proteins, as shown in Figure 5A. The error bars denote standard deviations over 10 runs, and statistical significance is confirmed with Wilcoxon signed-rank tests ($p < 0.01$). To further illustrate its behavior on individual examples, Figures 5B and 5C report cumulative rewards for the selected proteins. ThermoRL achieves performance comparable to or better than baseline methods in identifying stabilizing mutations, without retraining on target proteins. These results highlight ThermoRL's transferability across structurally diverse and previously unseen protein targets. In terms of efficiency, ThermoRL requires less than 2 seconds per mutation at inference, compared to 35–120 seconds for BO-based baselines, demonstrating clear scalability advantages.

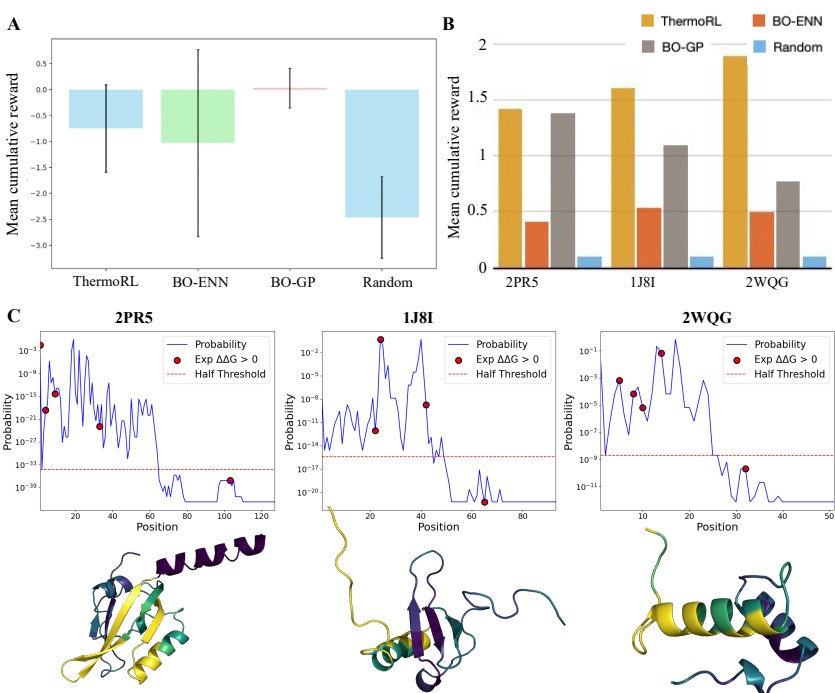

Figure 5: Evaluation of ThermoRL on unseen proteins. (A) Average cumulative reward across 200 unseen proteins comparing ThermoRL with BO-ENN, BO-GP, and Random Search. (B) Per-protein performance comparison on three representative test cases (2PRS, 1J8I, 2WQG), showing ThermoRL achieves competitive or superior results. (C) Probability profile and 3D structural visualization of mutation site prioritization by ThermoRL for each case. Plots (top) show position-wise selection probability with experimentally validated stabilizing mutations ($\Delta\Delta G > 0$) marked in red. Corresponding 3D structures (bottom) highlight high-probability regions in dark blue, which frequently coincide with functionally or structurally important residues.

**Visualization on Unseen Protein Cases** As shown in Figure 5B, the model computes a selection probability $P(position)$ for each residue in the protein sequence by applying a softmax over position-level rewards. This quantifies the relative importance of each site for thermostability optimization. By integrating these probabilities with three-dimensional structural representations (right panel), we visualized the spatial distribution of prioritized regions within each protein. In all three cases, the high-probability residues were concentrated in functionally or structurally critical regions, including active sites and hydrophobic cores. Notably, these prioritized regions showed significant overlap with experimentally validated stabilizing mutation sites ($\Delta\Delta G > 0$), indicating that ThermoRL is able to localize mutation targets with functional relevance. This demonstrates the framework's potential not only to guide efficient mutation planning but also to provide biologically meaningful design suggestions, supporting its application in structure-based protein engineering.

## 5 CONCLUSION

We introduced ThermoRL, a hierarchical reinforcement learning framework for protein thermostability optimization that incorporates structural information via GNN-based embeddings. ThermoRL employs a reward function to encourage exploration while reducing reliance on costly wet-lab experiments through surrogate evaluations in simulations. In contrast to exhaustive mutational scans that require retraining for each individual protein, our approach enables targeted and iterative mutation design through structure-informed decision-making. Experimental results demonstrate that ThermoRL successfully reduces the mutational search space while maintaining high transferability across diverse protein datasets. The broader implications of ThermoRL are demonstrated through its iterative decision-making mechanism. By integrating dynamic feedback, ThermoRL continuously refines mutation strategies, addressing the limitations of traditional one-step prediction approaches.

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

APPENDIX

# 6    DATASET ANALYSIS

## 6.1    DATA CLEANING

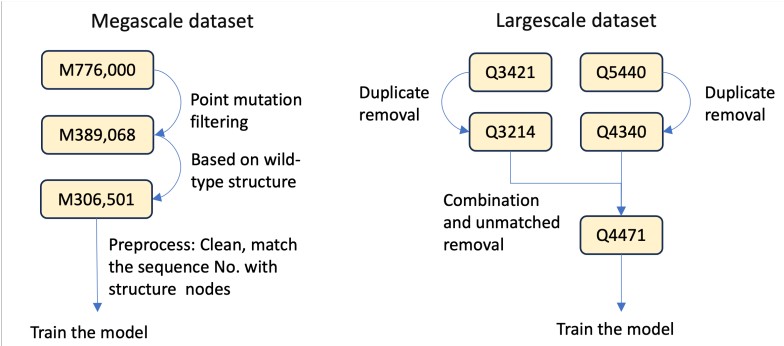

Figure 6: The cleaning process of the training Megascale and Largescale datasets.

The datasets used in this study include Megascale and Largescale for training, and Ssym and S669 for unseen testing. Megascale integrates high-throughput protease sensitivity experiments with low-throughput thermodynamic stability assays, focusing on small proteins (fewer than 75 residues) with dense mutation coverage, achieving up to 1,400 mutations per protein. For this study, data curation was performed by selecting only single point mutations with reliable $\Delta\Delta G$ values and valid wild-type structures to obtain a final dataset of 306,501 mutations across 337 proteins. Largescale, created by merging Q3421 and Q5440, with thermodynamic stability data gathered with more traditional biophysical techniques, underwent a rigorous cleaning process to remove duplicate and erroneous data. Data curation consisted of removing duplicates and points with missing information and unmatched sequences, then combined to form the final dataset, which consists of 4,471 mutations for 151 unique proteins. It spans a wider range of protein lengths but has sparse mutation coverage, with 50% of proteins containing fewer than 20 mutations. Figure 6 shows the cleaning process of these two training datasets. Figure 7 shows the performance of surrogate models on their corresponding training datasets, presenting the RMSE distribution across protein sequence lengths to highlight how surrogate model accuracy varies with protein size.

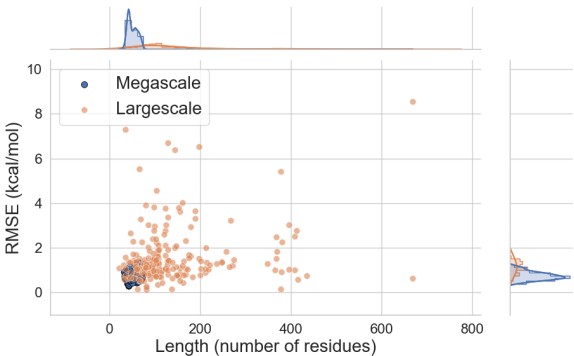

Figure 7: The RMSE distribution across protein sequence lengths of training datasets.

## 6.2    DIVERSITY ANALYSIS

The diversity of protein structures in a dataset is a critical factor influencing the performance and generalizability of machine learning models for protein engineering tasks. High structural diversity ensures the model learns robust representations, enabling better generalization to unseen proteins.

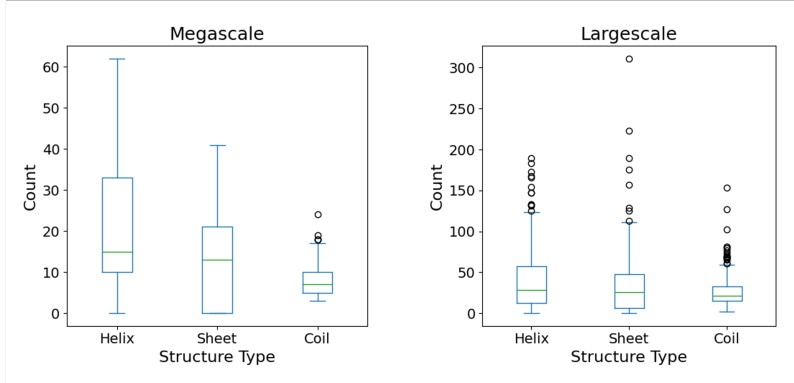

Figure 8: The distribution of secondary structures for protein strucutrues involved in Megascale and Largescale datasets.

To evaluate the structural diversity of the proteins in the Largescale and Megascale datasets, we performed the analysis focusing on secondary structure composition and overall structural variability.

In our analysis, we extracted secondary structure information from protein PDB files, focusing on the counts of $\alpha$-helices, $\beta$-sheets, and coils (unstructured regions). These structural features are fundamental building blocks of proteins, providing insight into their three-dimensional conformations. $\alpha$-helices are spiral-like structures stabilized by hydrogen bonds, contributing to the protein's flexibility and ability to bind to other molecules. $\beta$-sheets are flat, sheet-like structures that add rigidity and stability, often involved in structural support or protein-protein interactions. Coils represent unstructured regions that lack a defined secondary structure but are crucial for protein dynamics and adaptability, often mediating interactions or enabling conformational flexibility. By comparing the composition of these secondary structure elements across datasets, as shown in Figure 8, we aim to assess the diversity of protein structures. The secondary structure distribution in the Megascale dataset is more concentrated, with a smaller range, particularly in Helix and Sheet, where few notable outliers are observed. This indicates that the protein structural features in the Megascale dataset are relatively uniform. In contrast, the Largescale dataset shows a broader range across all secondary structure features and a greater number of outliers, with some proteins exhibiting significantly complex structures. This reflects the higher diversity of the Largescale dataset. Overall, the Megascale dataset is better suited for fine-grained mutation analysis or studies focused on specific protein families, while the Largescale dataset, due to its structural diversity, supports generalization studies for complex protein-related problems and offers broader application potential.

## 6.3 STRUCTURAL AND EMBEDDING-BASED SIMILARITY ANALYSIS

To assess dataset-level similarity, we compared distributional characteristics using both structural alignment and protein language model embeddings. The objective is to evaluate whether validation sequences originate from a distribution that is related, but not identical, to the training data.

To illustrate the correspondence between structure-based and embedding-based similarity, we compared representative protein pairs across datasets. For example, the proteins 2rn2 (from the unseen Ssym dataset) and 1l63 (from the Largescale dataset) yielded a TM-score of 0.26, indicating low structural similarity. This is consistent with the cosine similarity of 0.5412 obtained from ESM-2 embeddings. The TM-align results for this comparison are reproduced below:

```
*******************************************************************************
* TM-align (Version 20190822) *
* An algorithm for protein structure alignment and comparison *
* Based on statistics: *
* 0.0 < TM-score < 0.30, random structural similarity *
* 0.5 < TM-score < 1.00, in about the same fold *
* Reference: Y Zhang and J Skolnick, Nucl Acids Res 33, 2302-9 (2005) *
*******************************************************************************
```

```
Name of Chain_1: A287142
Name of Chain_2: B287142
Length of Chain_1: 162 residues
Length of Chain_2: 155 residues

Aligned length = 63, RMSD = 4.17, Seq_ID = 0.079
TM-score = 0.25675 (normalized by Chain_1 length)
TM-score = 0.26517 (normalized by Chain_2 length)
```

Further representative comparisons yielded TM-scores ranging from 0.2 to 0.7, corresponding to ESM-2 cosine similarities between approximately 0.5 and 0.9. These results indicate that protein language model embeddings capture aspects of structural similarity, despite not relying on explicit 3D information.

It is important to note that TM-score is inherently a pairwise metric. Applying it to dataset-level comparisons (e.g., 100 unseen proteins × 500 training proteins) would require up to 50,000 structural alignments. This approach is computationally prohibitive and does not yield a unified global similarity metric.

In contrast, embedding-based similarity using the pretrained ESM-2 model (facebook/esm2_t6_8M_UR50D) provides a scalable and consistent representation. ESM-2 embeddings encode both sequence-level and latent structural information in high-dimensional space, enabling efficient comparisons across datasets. This approach is aligned with recent practices in large-scale protein modeling and dataset characterization.

# 7 THERMORL

## 7.1 TRAINING ALGOTITHM

The surrogate model leverages a pre-trained graph encoder, initially trained in an unsupervised manner on a diverse set of protein structures. During surrogate training on task-specific datasets (thermostability $\Delta\Delta G$), the graph encoder was fine-tuned with a reduced learning rate to adapt to the regression task. Performance was evaluated using root-mean-square error ($RMSE$) and $R^2$, which measured prediction accuracy and model generalization, respectively. A robust 5-fold cross-validation strategy was employed to ensure reliable performance estimates by averaging results across multiple partitions. The fine-tuned graph encoder generates structural embeddings for the RL agent, facilitating efficient mutation optimization. The surrogate model serves as the reward signal in the RL framework.

## 7.2 ABLATION

We conducted ablation studies to evaluate the impact of graph encoder in the RL framework. Removing the structural graph encoder and replacing it with sequence-only embeddings resulted in reduced performance. This confirms that structural context is essential for generalizing across diverse protein folds.

# 8 COMPARISON OF SURROGATE MODELS

Hyperparameter tuning and ablation studies were performed on the Megascale dataset due to its compact size and dense mutation coverage, which provided a reliable platform for detecting subtle performance variations in the model. This choice ensured that the configurations and insights derived were robust and transferable, as demonstrated by the model's strong and consistent performance on the more diverse Largescale dataset.

---

**Algorithm 1** Training algorithm

---

1: **Input:** Wild type protein graph $G_i$, size $N$, training iteration $K$
2: Initialize $Q_\theta^1, Q_\phi^2$ with random weights
3: Initialize target networks $Q_\theta^{1,\text{target}}, Q_\phi^{2,\text{target}}$
4: Initialize history replay buffer $\mathcal{M}$;
5: Load pre-trained graph encoder $E_\psi$
6: Load pre-trained surrogate model $S_\xi$
7: **for** each training episode **do**
8:     Initialize state $s_0$
9:     **while** not terminal **do**
10:         Encode state using pre-trained graph encoder: $h_t = E_\psi(s_t)$
11:         Select first action $a_t^1$ using $\epsilon$-greedy policy:
12:             With probability $\epsilon$, select $a_t^1 \sim \text{Uniform}(V)$
13:             Otherwise, $a_t^1 = \arg\max_a Q_\theta^1(s_t, a)$
14:         Select second action $a_t^2$ hierarchically using $\epsilon$-greedy:
15:             With probability $\epsilon$, select $a_t^2 \sim \text{Uniform}(C')$
16:             Otherwise, $a_t^2 = \arg\max_a Q_\phi^2(s_t, a_t^1, a)$
17:         Predict reward $r_t$ using surrogate model: $r_t = S_\xi(h_t, a_t^1, a_t^2)$
18:         Observe next state $s_{t+1}$
19:         Store experience $(s_t, \{a_t^1, a_t^2\}, r_t, s_{t+1})$ in $\mathcal{M}$
20:         Sample mini-batch $\mathcal{B}$ from $\mathcal{M}$
21:         **for** each sample $(s, \{a^1, a^2\}, r, s') \in \mathcal{B}$ **do**
22:             Compute target for $Q^1$, Update $\theta$
23:             Compute target for $Q^2$, Update $\phi$.
24:         **end for**
25:         Periodically update target networks:
26:             $Q_\theta^{1,\text{target}} \leftarrow Q_\theta^1 , \qquad Q_\phi^{2,\text{target}} \leftarrow Q_\phi^2$
27:         Set $s_t = s_{t+1}$
28:     **end while**
29: **end for**

---

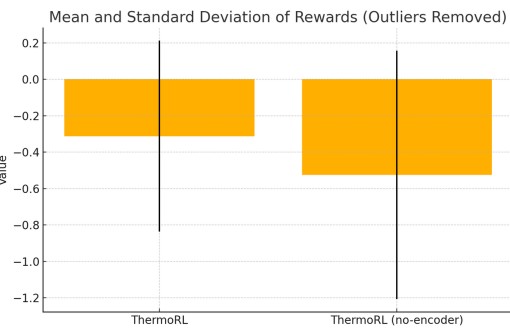

Figure 9: Ablation study on ThermoRL framework components.

## 8.1 COMPARISON WITH STRUCTURE-AWARE AND ENERGY-BASED BASELINES

We compared ThermoRL's surrogate models against a broader set of recent structure-aware and physics-based $\Delta\Delta G$ estimators, including RaSP, ProS-GNN, ThermoNet, Rosetta, FoldX, ThermoMPNN, and ACDCNN. Results are shown on the widely used Ssym and S669 benchmark datasets.

Regarding DeepDDG, we acknowledge its importance as a sequence-based predictor without 3D structure. However, the original version (2019) relied on PSI-BLAST PSSM acceleration and is an ensemble-based method without public code, limiting reproducibility. As a result, most subsequent works only cite it rather than conducting direct large-scale comparisons.

| Model | RMSE (Ssym) | PCC (Ssym) |
|---|---|---|
| LargeSurrogate | 1.31 | 0.57 |
| MegaSurrogate | 1.49 | 0.14 |
| ProteinMPNN | 3.20 | 0.56 |
| ThermoMPNN | 1.53 | 0.60 |
| ThermoNet | 1.55 | 0.47 |
| ACDCNN | 1.47 | 0.55 |
| RaSP | 1.97 | 0.23 |
| ProS-GNN | 1.53 | 0.05 |
| Rosetta | 2.61 | 0.43 |
| FoldX | 2.13 | 0.39 |

Table 1: Performance on Ssym dataset across multiple baselines.

| Model | RMSE (S669) | PCC (S669) |
|---|---|---|
| LargeSurrogate | 1.60 | 0.47 |
| MegaSurrogate | 1.73 | 0.41 |
| ProteinMPNN | 3.32 | 0.26 |
| ThermoMPNN | 1.52 | 0.43 |
| ThermoNet | 1.62 | 0.39 |
| ACDCNN | 1.60 | 0.46 |
| RaSP | 1.63 | 0.39 |
| ProS-GNN | 1.70 | 0.13 |
| Rosetta | 2.70 | 0.39 |
| FoldX | 2.30 | 0.22 |

Table 2: Performance on S669 dataset across multiple baselines.

## 8.2 ABLATION

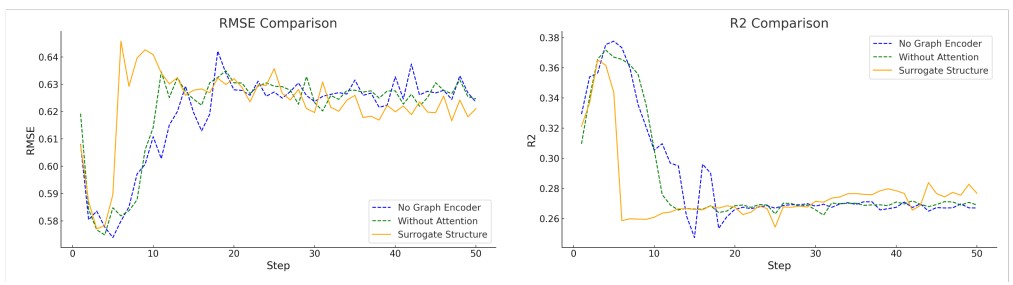

Figure 10: Ablation Study: Impact of Removing Different Modules on RMSE and R² Performance.

This set of comparison plots illustrates the results of two ablation experiments: the blue dashed line ("No Graph Encoder") represents the removal of the Graph Encoder module, while the green dashed line ("Without Attention") reflects replacing the Attention Interaction module with a direct concatenation of Wild Type GNN Embeddings and Difference Graph Embeddings. The results show that removing or modifying either module leads to a performance decline, with the original structure (orange solid line) demonstrating the best performance, highlighting its superior and effective design.

## 8.3 HYPERPARAMETERS

This figure11 summarizes the results of hyperparameter tuning experiments, highlighting the impact of batch sizes, GNN depth, and dropout rates on model performance, as measured by RMSE and R² across training steps. The overall observation is that the influence of these hyperparameters on performance is relatively modest, with differences between configurations being subtle. For batch sizes, the variations across 16, 32, and 64 indicate that smaller batch sizes, particularly 32 (orange line), consistently provide a slight performance advantage. For GNN depth, Depth 3 (orange line)

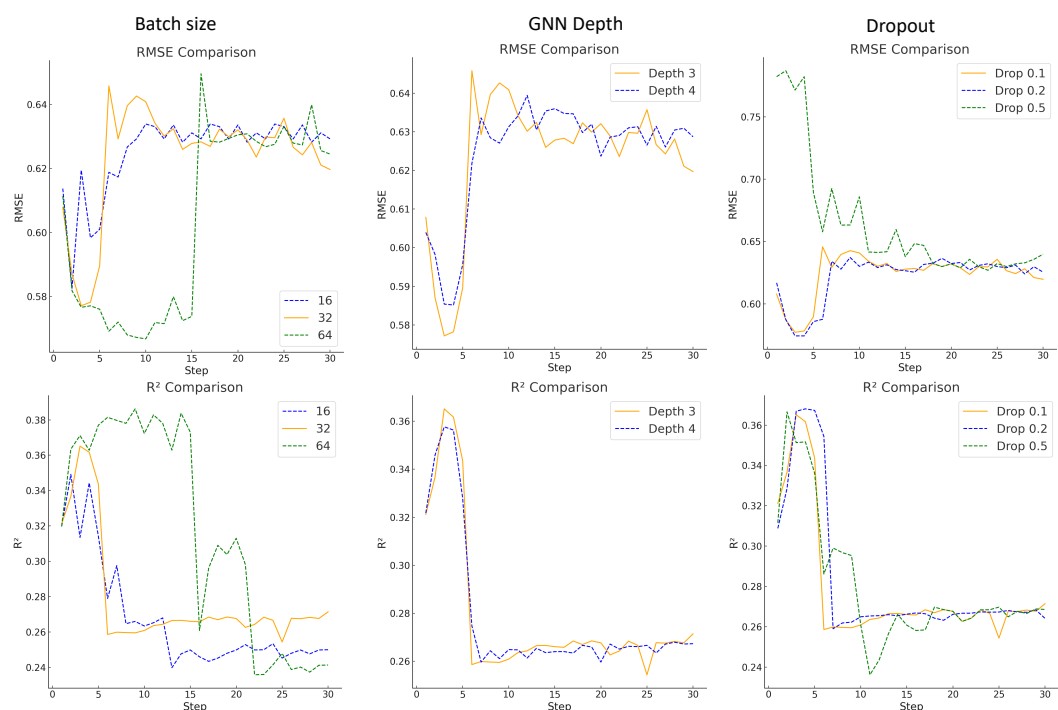

Figure 11: Hyperparameter Tuning: Impact of Batch Sizes, GNN Depth, Dropout Rates on RMSE and $R^2$

outperforms Depth 4 in terms of stability and accuracy, though both configurations yield comparable results overall. Similarly, for dropout rates, the model with Dropout 0.1 (orange line) delivers the best performance by balancing regularization and retention of model complexity effectively. Overall, while the differences are not drastic, the orange lines across all plots represent the selected best-performing models for each hyperparameter configuration. These hyperparameters were subsequently used in the final model, ensuring a balanced and optimal outcome.

