# OpenReview forum: "ThermoRL: Structure-Aware RL for Protein Mutation Design to Enhance Thermostability"
_ICLR.cc/2026/Conference — ICLR 2026 Conference Withdrawn Submission_

### Official Review · Reviewer_1Y6p · 2025-10-24

**Soundness:** 2
**Presentation:** 1
**Contribution:** 2
**Rating:** 2
**Confidence:** 3

**Summary:**

The paper presents ThermoRL, a structure-aware hierarchical DQN framework for sequential protein thermostability design. The key idea is to decompose a mutation decision into two steps—choose a residue position, then choose its amino-acid substitution—reducing the action space and improving exploration efficiency. The reward is provided by a surrogate ∆∆G model that integrates a wild-type structural graph with a difference graph constructed from a predicted mutant structure (via PreMut) and uses cross-attention to map mutation features to wild-type embeddings; the surrogate then outputs ∆∆G used as the immediate reward. Empirically, the authors evaluate 200 unseen proteins, comparing ThermoRL with BO-GP, BO-ENN, and Random. ThermoRL is competitive or better, with <2s per mutation inference vs 35–120s for BO baselines.

**Strengths:**

* Sound problem decomposition. The hierarchical action design is principled and clearly reduces complexity, which matters in protein-scale action spaces.

* Reward modeling with structural signal. The surrogate fuses a wild-type graph and a mutant difference graph with cross-attention at residue granularity—a thoughtful design that should capture local interaction changes.

**Weaknesses:**

Despite ThermoRL having reached good performance, I still have several concerns:
* Reward-model bias. Because the policy optimizes the surrogate rather than the ground truth, it risks overfitting to surrogate idiosyncrasies. This increases the risk that ThermoRL is fitted to the surrogate model rather than to real mutation effects.
* Multi-mutation policies. The text mentions extending to multi-step design and thresholded stopping, but experiments focus on single-step choices; it would be useful to include 2–4 site designs to test epistasis handling and stopping rules. Moreover, the claimed contribution "Trajectory-based mutation planning" is somewhat confusing. To my understanding, the mutation trajectory involves multiple cumulative mutation effects, but the experiments do not reflect this.
* Structural awareness. ThermoRL claims to be more effective at leveraging structural context, but for now, this is reflected only in building edges with a cutoff of 8 Å. If 3D structural information is used in a different manner, please specify how.
* Limited technical improvements. The core modules are GNN, RL (DQN), which are nothing new.
* Vague figures. Fig. 4A and Fig.5B can be improved.
* No code implementation provided.

**Questions:**

* What is the performance of ThermoRL on all proteins in Figure 4A? Why select these specific proteins?
* Is there any latest progress in this field, considering the baseline (BO) was published in 2020?
* Can you provide detailed numerical results of Figure 5A? It seems BO-GP is a better method.

---

### Official Review · Reviewer_n448 · 2025-10-29

**Soundness:** 1
**Presentation:** 2
**Contribution:** 3
**Rating:** 2
**Confidence:** 4

**Summary:**

Authors introduce ThermoRL, a reinforcement learning framework that integrates a GNN-based encoder with hierarchical Q-learning to design thermostabilizing protein mutations sequentially.

**Strengths:**

1. The paper argues two limitations of existing mutation optimization methods. 1) Inefficient selection of mutation site - Random or predict-then-rank. 2) Limited use of structural context. These two limitations are very accurate and critical, making the motivation of the paper strong.
2. Discussion on surrogate model (4.1) is robust and reasonable. For benchmarks that uses surrogate as evaluation model, it’s important to discuss the accuracy and generalizability of the surrogate, which authors do in section 4.1, App 6.2-6.3.
3. ThermoRL is not trained to individual protein family, it is a general method that can introduce stabilizing mutations to any protein.

**Weaknesses:**

1. **Baselines are very weak.** Only random baseline and two very vanilla BayesOpt. I strongly disagree to Line 353-357 ““predict-then-rank” manner over fixed candidate sets. These models do not perform decision-making or mutation selection, and are thus not directly comparable to ThermoRL. We include them here solely for benchmarking ∆∆Gprediction accuracy. Therefore, they are not directly comparable to ThermoRL”. **I find this argument unconvincing. There are many reasonable baselines the authors could have implemented.**
  - I strongly disagree that predict-then-rank methods aren’t directly comparable to ThermoRL. It’s straightforward to make them comparable by using these models as selection mechanisms within an evolutionary algorithm or as the acquisition function in BayesOpt. Even more simply, one could generate a reasonable number of candidate sequences through random mutation and screen them.
  - Why not compare against latent BO methods using the same graph encoder embeddings that ThermoRL employs? There’s not even a discussion of latent BO for proteins. For reference, see Accelerating Bayesian Optimization for Biological Sequence Design with Denoising Autoencoders (ICML 2022).

2. **Missing discussion of important recent works.** Continuing from the point above, the paper overlooks several relevant and recent works on protein fitness optimization. I understand the authors’ setting is different (a general method applicable to any protein vs. methods trained to optimize a specific protein), but conceptual comparisons would still be valuable.
- Improving Protein Optimization with Smoothed Fitness Landscapes (ICLR 2024): worth mentioning, as this work does not have the predict-then-rank limitation and samples mutations via gradient-based updates.
- Robust Optimization in Protein Fitness Landscapes Using Reinforcement Learning in Latent Space (ICML 2024): a more recent and better-performing RL approach than DynaPPO (line 147).

3. It’s standard in RL literature to plot cumulative reward (training curve) as a continuous curve over time (x-axis as steps). That would allow a clearer comparison of optimization efficiency across methods.

4. **Missing analysis of optimized sequences and trajectories.** There’s no discussion of the optimized sequences or mutation trajectories. How long are these trajectories (i.e., how many mutations per protein)? The authors mention running 10 runs per model. But how diverse are the resulting sequences?

5. Minor: In Fig. 3C, the x/y-axis tick labels are too small; it’s hard to read the numbers.

Overall, the evaluation and experimental execution feel weak. I would be happy to raise my score if the authors address these issues.

**Questions:**

1. Appendix 8.2-8.3: What exactly does the x-axis step represent? Is it a validation step or a training step? How does it map to the training iterations? If 30 steps literally means 30 backpropagations, I don’t think this serves as a meaningful ablation.
2. State space: What is the input to the graph encoder? My understanding is that node features represent the current sequence, while edge features (lines 185–187) are derived from the wild-type structure. Since there’s no mention of recomputing structure using a prediction model, what happens when a substantial number of mutations alter the structure?

---

### Official Review · Reviewer_wkXi · 2025-10-31

**Soundness:** 2
**Presentation:** 2
**Contribution:** 2
**Rating:** 2
**Confidence:** 4

**Summary:**

The authors propose ThermoRL, a framework for thermostability optimization that uses a surrogate model based on a GNN for DDG prediction and a hierarchical RL policy that chooses a position and an amino acid for that position, given an input sequence and structure for a wild-type protein. Results show that their surrogate model achieves state-of-the-art performance for DDG prediction and that the RL policy is effective for suggesting mutations.

**Strengths:**

1. The authors propose a GNN-based architecture for DDG prediction, achieving state-of-the-art results.
2. Results show that the RL policy assigns a higher probability to positions near the functional motifs of target proteins.
3. The proposed hierarchical formulation can lead to enhanced computational efficiency.

**Weaknesses:**

1. Information regarding RL policy, hyperparameters (like terminal state), seems to be missing to reproduce this work.
2. The baselines chosen for comparison are very simple, and there is no information on how the BO baselines are implemented.
3. The modeling of the RL framework is weak, needing additional justification for choices.
4. Code is not available.

**Questions:**

I initially recommend rejection, with the following reasons: (i) lack of critical information to reproduce this work, (ii) need to add stronger baselines for comparison, and (iii) clarification needed regarding RL formulation. My detailed comments are as follows.

Comments:

1. Citing and discussing the work for stability prediction presented in Cagiada et al “Predicting absolute protein folding stability using generative models” is missing. They also evaluate structure-aware models, like ESM-IF, for stability prediction.
2. (lines 315-316) For the datasets evaluated, do all the wild-types have experimental structures?
3. What is the exact architecture of the RL policy?
4. Information on hyperparameters, optimizers seems to be missing, e.g., does the information in lines 242-243 mean that each episode consists only of one action for position and mutation?
5. (lines 361-366) Can the authors clarify if this joint probability was used for reward calculation, performance evaluation, or both?
6. The authors mention in lines 401-403 that BO and RS methods are specifically designed for individual proteins, but no information regarding BO training is given, and RS is just random search.
7. (lines 840-842) What is the loss function when training the graph encoder in an unsupervised manner?
8. The evaluation protocol and the baselines chosen are very weak; at least the following comparisons seem needed: (i) comparing with a methodology that uses ESM-IF to find positions to mutate and mutations for those positions, (ii) showing evaluations in Fig. 4 not only to the proposed method but also for other surrogates presented in the Appendix.
9. Given the need of stochasticity in selecting mutations, why did the authors choose to apply a softmax in the output of a Q-learning-based policy instead of using another RL algorithm?

Minor Comments (that did not impact the score):

1. The use of \citet and \citep should be revised in the entire manuscript for references.
2. Fig. 1 could be improved by explicitly showing that the proposed method uses structural information.
3. The explanation of some methods in the Related Work section needs improvement.
4. Typos: (line 198) “Finite”, (line 267) “fist”
5. Readability of the axis in Fig. 3 can be improved.

---

### Note · Authors · 2025-11-12

I have read and agree with the venue's withdrawal policy on behalf of myself and my co-authors.